# Modelling the impact of vaccine hesitancy in prolonging the need for Non-Pharmaceutical Interventions to control the COVID-19 pandemic

Daniela Olivera Mesa [1✉], Alexandra B. Hogan [1], Oliver J. Watson [1], Giovanni D. Charles[1], Katharina Hauck[1], Azra C. Ghani[1] & Peter Winskill [1]

**Abstract**

**Background** Vaccine hesitancy – a delay in acceptance or refusal of vaccines despite availability – has the potential to threaten the successful roll-out of SARS-CoV-2 vaccines globally. In this study, we aim to understand the likely impact of vaccine hesitancy on the control of the COVID-19 pandemic.

**Methods** We modelled the potential impact of vaccine hesitancy on the control of the pandemic and the relaxation of non-pharmaceutical interventions (NPIs) by combining an epidemiological model of SARS-CoV-2 transmission with data on vaccine hesitancy from population surveys.

**Results** Our simulations suggest that the mortality over a 2-year period could be up to 7.6 times higher in countries with high vaccine hesitancy compared to an ideal vaccination uptake if NPIs are relaxed. Alternatively, high vaccine hesitancy could prolong the need for NPIs to remain in place.

**Conclusions** While vaccination is an individual choice, vaccine-hesitant individuals have a substantial impact on the pandemic trajectory, which may challenge current efforts to control COVID-19. In order to prevent such outcomes, addressing vaccine hesitancy with behavioural interventions is an important priority in the control of the COVID-19 pandemic.

**Plain language summary**

People refusing or delaying COVID-19 vaccination might impact current efforts to control the pandemic caused by SARS-CoV-2. Here, we have examined the effects of low vaccine uptake due to vaccine hesitancy on the need to prolong other public health measures to control the pandemic. We used mathematical modelling and data on vaccine hesitancy from population surveys across different countries. Our results suggest that when there is vaccine hesitancy and relaxation of other public health measures, mortality could increase by up to seven times compared with ideal vaccination coverage of the population. Furthermore, for some scenarios analysed, longer and more stringent public health measures would be required to compensate for lower vaccine uptake. Our work demonstrates that vaccine hesitancy might have a substantial health impact on the population, and therefore, it is a public health priority to increase trust in vaccines.

[1] MRC Centre for Global Infectious Disease Analysis; and the Jameel Institute, School of Public Health, Imperial College London, London, UK.
✉email: d.olivera-mesa17@imperial.ac.uk

The COVID-19 pandemic has simultaneously resulted in high global mortality and major economic disruptions. As a control measure, non-pharmaceutical interventions (NPIs) such as social distancing and mobility restrictions have been put in place worldwide and have successfully reduced transmission of the virus. However, these interventions are unsustainable in the long-term[1] and current hopes to control the pandemic rely heavily on vaccination.

In December 2020, the first vaccine against SARS-CoV-2 was approved; by May 2021, 14 vaccines had been licensed (https://vac-lshtm.shinyapps.io/ncov_vaccine_landscape/) and more than 1.3 billion vaccination doses administered worldwide (https://ourworldindata.org/covid-vaccinations#). Their reported efficacy against symptomatic disease ranges from 50% to over 95%[2–6]. Given the high basic reproduction number for SARS-CoV-2 (estimates range between 3–4)[1] high levels of vaccine uptake will be required to achieve herd immunity[7], particularly if children are not vaccinated during the first phase of roll-out.

One major concern that threatens to limit the impact of vaccination is vaccine hesitancy[8]. Population surveys have found that between 14%[9] and 27%[10] of adults say that they will not accept a vaccine if available, whilst between 14%[9] and 19%[10] say that they are uncertain. There is a large variation in vaccine hesitancy between countries, with the proportion saying that they would get a SARS-Cov-2 vaccine if it became available, ranging from 40% for France[10] to 89% for China[9]. In many countries, vaccine hesitancy is heterogenous across sub-populations depending on gender, age, ethnicity, religion, or socioeconomic status[9–11]. Surveys have highlighted the key drivers of SARS-CoV-2 vaccine hesitancy are related to concerns about the accelerated pace of vaccine development[11], side-effects[10] and the spread of misinformation about the pandemic[8]. Underlying reasons of vaccine hesitancy are a complex interaction between trust in government and health authorities[9] coupled with new information —and misinformation— on the vaccine safety and disease risk arising everyday[12].

In the present study, we aim to understand the likely impact of vaccine hesitancy on future control of the pandemic, using a mathematical model of SARS-CoV-2 transmission[7] to explore vaccine hesitancy through its impact on population coverage. We capture the effect of reduced coverage using measured levels of vaccine hesitancy from behavioural survey data[10] on self-reported intention to be vaccinated. Survey results are disaggregated by age and translated to vaccination coverage ranges per age group. Pandemic trajectories with low vaccination coverage due to vaccine hesitancy are compared to an ideal counterfactual assuming no vaccine hesitancy, in which we assume that a small proportion (5%) of the population cannot be reached for vaccination. This value is based on maximum vaccination uptake reported for England's current COVID-19 vaccine rollout (https://www.england.nhs.uk/statistics/statistical-work-areas/covid-19-vaccinations/). We model each scenario with both a high and a moderate vaccine efficacy profile that represents the range of efficacies of currently approved vaccines. Informed by current vaccine roll-out in high-income countries, we assume that vaccination started in January 2021 and is implemented at a rate that results in a total campaign of 10 months to fully vaccinate the population above 15 years old.

Our simulations suggest that mortality could be higher in countries with high vaccine hesitancy compared to an ideal vaccination and this could prolong the need for NPIs to remain in place. We show that to reduce this impact, vaccination campaigns could include less vulnerable groups, like children. Vaccine hesitancy is an important public health priority that needs to be addressed in order to control the current pandemic.

## Methods

**Vaccine hesitancy data.** Attitudes towards COVID-19 vaccination were obtained from the Imperial College London YouGov Covid 19 Behaviour Tracker Data[10]. This data set includes weekly surveys about people's behaviours in response to COVID-19 (including vaccines) as well as standard demographic questions on age, gender and household structure. Ethics approval and informed consent were not required given that all data was publicly available and de-identified.

We extracted the survey results from 8th to 15th February 2021 for 10 European countries. To assess vaccine hesitancy, we used data from one question pertaining to COVID-19 vaccine acceptance in which participants were asked to what extent they would definitely get a COVID-19 vaccine, if it became available to them next week. Answers were obtained on a numeric scale ranging from "Strongly agree – 1" to "Strongly disagree – 5". To capture survey uncertainty, answers per age group were used to parameterise a multinomial distribution, from which we drew 100 replicates. To capture further uncertainty associated with the translation of survey response to vaccine uptake, for each replicate, coverage per age group was estimated assuming the probability of vaccination as a beta distribution with means: 0.98, 0.75, 0.50, 0.25 and 0.02 for survey responses 1, 2, 3, 4 and 5, respectively. Coverage distributions per age group, median as well as the 10% and 90% quantiles are shown in Table S5 and Fig. S2.

**Mathematical model.** We used a previously developed mathematical model for SARS-CoV-2 transmission and vaccination[7] (Fig. S1). The age-structured deterministic SEIR-type compartmental model incorporates an age-specific probability of infection determined by age-based contact matrices. Susceptible individuals become infected at a rate that depends on the level of infection in the community. Following infection, cases proceed to mild infection or a clinical disease pathway, which includes hospitalisation, oxygen support and intensive care. Waning immunity is captured by recovered individuals returning to the susceptible compartment following an erlang distribution.

Vaccination is modelled as an additional dimension disaggregating the population into those who have not received the vaccine (v0), those who have received the vaccine but are not yet protected (this stage represents the two-dose vaccine schedule and the need to wait ~28 days from dose 1 for protection to develop) (v1 and v2), those who have received the vaccine and are protected (v3 and v4) and those who have received the vaccine but are no-longer protected (v5) (if vaccine-derived immunity is not life-long). In this model, only those who are currently infected do not receive the vaccine. Protection due to vaccination is modelled at two stages in the model; (1) reducing the probability of infection upon exposure (efficacy against infection) and (2) reducing the probability of hospitalisation being indicated after developing disease (efficacy against hospitalisation and death).

**Parameters.** Parameters for SARS-CoV-2 infection, health care capacity, age-distribution and contact patterns are based on previous work[7,13] (Tables S1 and S4). Given these parameters, transmission probability is estimated based on reproductive number (Rt), which is given as an input for each simulation as a function of time. Vaccine-induced immunity was assumed lifelong, while natural immunity was assumed to last for an average of one year[14]. To produce simulations representing the different vaccines approved to date, each scenario was run for two vaccines: one with high efficacy (94% efficacy against infection[2]) and one with moderate efficacy (63% efficacy against infection[3]). For both vaccines we assume an additional 60% efficacy against hospitalisation for breakthrough infections, resulting in an overall vaccine efficacy against hospitalisation and death of 98% for

the high efficacy vaccine and 85% for the moderate efficacy vaccine. A summary of key parameters is given in Tables S1–S6. The model code is freely available at https://github.com/mrc-ide/nimue[15].

To mimic current vaccine rollout plans, vaccination is introduced in the population at the beginning of January 2021. We assumed a constant vaccination rate ($\kappa$), at which all individuals aged 15 years and above (~78% of the population) will be vaccinated over a 10-month period. This rate is implemented for all scenarios modelled, since we assume vaccination rate is constrained not by vaccine uptake but by the supply and delivery of vaccines. Therefore, lower levels of coverage, result in shorter vaccination campaigns; given that in the model, once coverage targets are met, vaccination is ceased. To illustrate the effect including children vaccination, vaccination rate was maintained constant and vaccination period was extended such that all individuals age 5–15 years could be vaccinated.

Vaccines are targeted by age groups at the constant rate $\kappa$, prioritising older age groups: with 80+ years vaccinated first and then sequentially including additional age groups in 5-year age-bands down to 15–19 years for adults only vaccination simulations and down to 5–10 years for simulations including children vaccination.

**Reproductive number profiles**. To simulate a representative pre-vaccination scenario, we generated a reproductive number profile in which Rt was the same as $R_0$ ($R_0 = 3$[13]) up to April 2020, subsequently decreased to 1 to represent the impact of NPIs against the first wave, and then rose to 1.5 during the latter half of 2020 to represent a second wave. Following the introduction of vaccination in January 2021, we set Rt to increase in 10 fixed steps. Each step representing the lifting of NPIs. The time for each step increase was determined by estimating when vaccination coverage had reached levels such that the herd immunity threshold due to vaccine immunity was reached. At the end of the vaccination period, $R_t$ remained at a value such that the herd immunity threshold was maintained, given final vaccination coverage and vaccine efficacy against infection.

To estimate the coverage needed for each Rt step, the following herd immunity threshold equation was used:

$$\text{Coverage} = \left(1 - \frac{1}{R_t}\right)\frac{1}{\text{efficacy}}$$

When analysing the impact of lifting NPIs, the Rt profile following the introduction of vaccination was generated based on an ideal scenario for vaccination uptake. Conversely, when evaluating the degree to which NPIs would need to remain in place, the Rt profile after the introduction of vaccination was set up based on vaccine coverage due to vaccine hesitancy.

**Scenarios**. We consider two potential scenarios for vaccine coverage target per age group: an ideal scenario 2020 with 20 cases. A simulation was run for each vaccine coverage scenario for both adult-only vaccination campaign and vaccination campaign including children. As an output for each simulation, we estimated the number of deaths and hospitalisations associated with COVID-19 over the 2-year period from 1 January 2021 to 31 December 2022.

To generate country-specific simulations, we parameterise the model with data on the population size and age distribution of the country (https://population.un.org/wpp/) and representative contact matrices obtained from a systematic review of social contact surveys through the socialmixR package (https://github.com/sbfnk/socialmixr). The model was then fitted to reported daily cases and deaths up to 31 December 2020 by varying three parameters - the start date of the epidemic, the initial R0 and the

effect size of changes in mobility on transmission (using mobility data from Google (https://www.google.com/covid19/mobility)). Model fitting was performed using a Metropolis Hastings MCMC based sampling scheme as previously described[16]. The resulting fit generates a fitted R0 as baseline, an Rt trajectory up to the introduction vaccination in January 2021, after which, Rt was set to increase by 10 fixed steps, up to the theoretical herd immunity threshold based on an ideal vaccination schedule (as described above). The pandemic trajectory was evaluated using country-specific data on vaccine hesitancy and demography for the two coverage scenarios described above and assuming vaccination for individuals aged 15 years and above only.

**Reporting summary**. Further information on research design is available in the Nature Research Reporting Summary linked to this article.

## Results

**Vaccine hesitancy public health impact**. We first sought to determine the public health impact of vaccination and vaccine hesitancy as NPIs are lifted. To do so, we allowed the time-varying reproductive number in the absence of immunity $R_t$, to be increased in steps such that the herd immunity threshold accounting for vaccine-induced immunity was maintained, under the assumption of ideal vaccination uptake (Fig. 1a, c). In this ideal scenario, NPIs can be fully lifted at the end of the vaccination period with a high efficacy vaccine (94% efficacy, Fig. 1a). However, with a moderate efficacy vaccine (63% efficacy), some NPIs or other population-level behavioural changes may need to remain to control the epidemic (Fig. 1c).

In the presence of vaccine hesitancy, lifting NPIs and relying on vaccine-induced immunity for control is predicted to lead to periodic outbreaks determined by the duration of naturally induced immunity (Fig. 1b, d). For a high efficacy vaccine, daily deaths per million at the peak of the first outbreak are projected to be 11.5 (10.1–13.2) times higher than under the ideal scenario (Fig. 1b). This translates to a cumulative impact of 532 (457–612) more deaths per million population in the two years after vaccination begins. In our results, fewer deaths are projected for a vaccine of moderate efficacy compared to a higher efficacy vaccine. This is partly due prolonged NPIs being required to maintain herd immunity where efficacy is lower, resulting in an outbreak that is more spread out and resulting in a lower final Rt compared to the high vaccine efficacy simulations. For a moderate efficacy vaccine, the cumulative impact of vaccine hesitancy is projected to lead to 456 (416–504) extra deaths per million population.

These adverse impacts of vaccine hesitancy on transmission, symptomatic disease, hospitalisations and deaths affect vaccinated as well as unvaccinated individuals because of imperfect vaccine efficacy (Fig. 2). Under the vaccine hesitancy scenario, the resulting lower vaccination coverage is projected to lead to a 16.7% and 30.4% increase in hospitalisations in the vaccinated population for the high and moderate vaccine efficacy profile, respectively, and a 9.4% and 27.2% increase in deaths in the vaccinated population, compared to an ideal vaccination scenario (Fig. 2).

**Relaxation of NPIs**. As an alternative way to assess the impact of vaccine hesitancy on the pandemic, we evaluated the degree to which other NPIs would need to remain in place given the real-time achieved vaccine coverage in order to prevent further epidemics (i.e. maintain herd immunity threshold, Fig. 3). For the high efficacy vaccine, under the ideal scenario, we predict that NPIs could be fully lifted by the end of 2021 whilst keeping transmission under control (Fig. S3). However, under the vaccine hesitancy scenario, limited NPIs or other behavioural

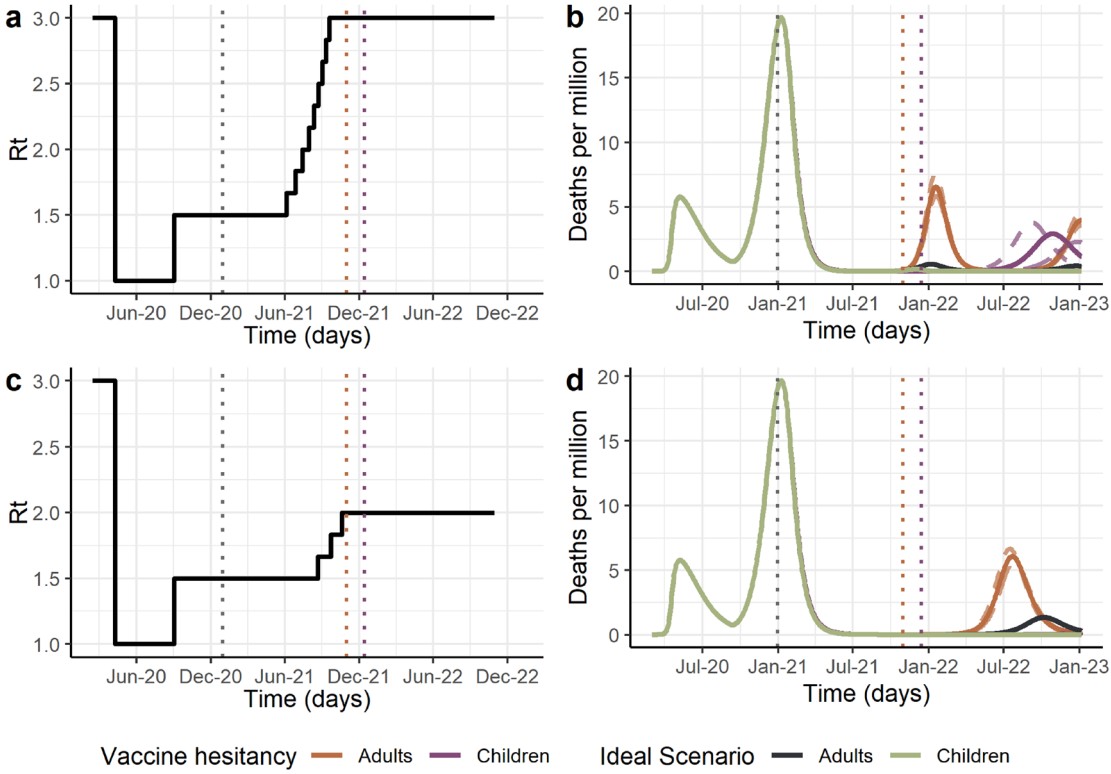

**Fig. 1 Projected COVID-19 dynamics given vaccine hesitancy.** Panels **a**, **b** show a high vaccine efficacy (94% against infection, 98% against hospitalisation and death), panels **c**, **d** moderate vaccine efficacy (63% against infection, 85% against hospitalisation and death). Panels **a** and **c** show the reproductive number Rt profile, which represents the level of NPI stringency, with lower numbers indicating higher stringency. In this illustrative example, we assume that a first wave of transmission occurred at the beginning of 2020 with the assumed value of Ro: 3. This was followed by NPIs leading to a reduction in Rt to 1, followed by an Rt of 1.5 as NPIs are lifted leading to a second wave of transmission in the latter half of 2020. After vaccination is introduced at the beginning of 2021, NPIs in all scenarios are lifted according to a schedule based on coverage under the ideal scenario (no vaccine hesitancy, 95% of individuals 15 years plus are vaccinated). Panels **b** and **d** show projected deaths per million under vaccine hesitancy scenarios: adults-only vaccination (orange), vaccination including children (purple). Continuous lines represent simulations of median vaccine coverage per age group, while dashed lines represent simulation of 10% and 90% quantiles. For the ideal scenario black line represents adults-only vaccination and green line represents ideal scenario when children vaccination is considered. In each scenario, final vaccination coverage per age group and deaths vary according to vaccine hesitancy. Vertical dashed lines indicate the vaccination rollout period in the ideal scenario.

modifications might need to remain in place, with Rt having to stay below 2.05 (1.96–2.14) to prevent further epidemics, this represents a 32% reduction of the assumed R0 of 3. A difference of ~35% in the effective reproductive number could represent the closure of educational institutions or limiting interaction between households to achieve control of the epidemic;[17] both of which are not sustainable or desirable.

**Vaccination of children.** As current vaccination rollout plan of adults continues swiftly in most high-income countries, public health authorities are now looking to include children into their vaccination campaigns while results of COVID-19 vaccine efficacy in children become available[18]. To evaluate the impact of including children in vaccination rollouts, we model all scenarios with a longer vaccination campaign, which allowed individuals above 5 years old to get vaccinated, assuming vaccine hesitancy for 5–17 years old the same levels reported for 18–24 years old[10]. If children are included in vaccine rollout, our results illustrate that in a scenario with vaccine hesitancy daily deaths per million at the peak of the first outbreak could be reduced by 56% (51–60%) for a vaccine with high efficacy (Fig. 1b). Which implies a total reduction of 272 (242–346) deaths per million in the two

years after vaccination begins (Fig. S4). For a moderate vaccine efficacy, higher NPIs stringency at the end of vaccine rollout entails later outbreaks, which do not take place during the two years after vaccination begins, resulting in similar results for the ideal and vaccine hesitancy scenario when including the vaccination of children (Fig. 1d and S4). Including children in vaccine rollout leads to higher vaccine coverage that compensates for vaccine hesitancy levels in adults. This is evident when evaluating the degree to which other NPIs would need to remain in place in order to maintain the herd immunity threshold based on vaccine-acquired immunity levels. For a high efficacy vaccine, in a vaccine hesitancy scenario Rt levels can increase up to 2.5 (Fig. 3b), ~20% more than for adult-only vaccination rollout. This increase entails milder NPIs at the end of vaccination campaign.

**Country-specific simulations.** Our illustrative examples above are comparable to the waves of COVID-19 outbreaks in Europe. However, vaccine hesitancy varies between countries. To evaluate the impact of these variations, we chose three European countries with different vaccine acceptance views: France, Germany and the United Kingdom (UK) (Fig. 4b). For each country, we fit the pandemic trajectory to country-specific data up to vaccination

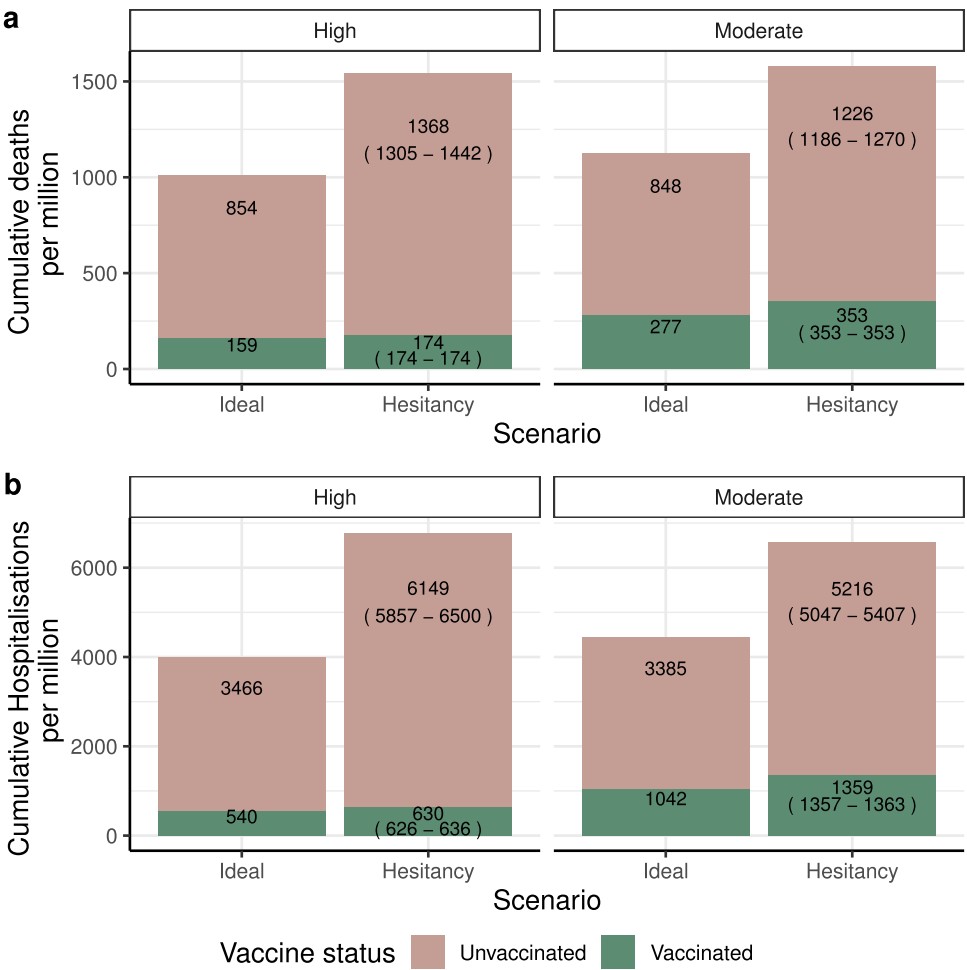

**Fig. 2 Public health impact of vaccine hesitancy.** High vaccine efficacy is shown on the left and moderate vaccine efficacy on the right. The annotated numbers are the cumulative deaths (**a**) and hospitalisations (**b**) per million individuals for the vaccinated and unvaccinated populations at the end of the projection horizon (1 January 2021–31 December 2022). Vaccination coverage of individuals aged 15 years and older is highest in the ideal scenario at 95%. For the hesitancy scenario annotated number is for median vaccine coverage per age groups, number in parenthesis are results for 10% and 90% quantiles coverage per age group.

started (1 January 2021), after which we model the trajectory of the pandemic under an ideal vaccination and a vaccine hesitancy scenario for each country independently (Fig. 4c)

For a vaccine with high efficacy, we project 1.2 (1.1–1.3), 5.0 (4.0- 6.3)- and 6.6 (5.7–7.6) times more deaths in 2021/2022 in a scenario with hesitancy compared to an ideal scenario in the UK, Germany and France respectively (Fig. 4a Death ratios vary between age groups, vaccine efficacy and countries depending on deaths predicted in their corresponding ideal scenarios. Nonetheless, for both high and moderate vaccine efficacy, the highest impact on total deaths is for the oldest age groups and it increases in countries with higher vaccine hesitancy (Figs. S5 and S6).

**Discussion**

We have examined the effects of low vaccine uptake due to vaccine hesitancy for the current COVID-19 pandemic and have shown the impact of vaccine hesitancy, detailing the considerable mortality that could be averted with increased vaccine coverage. Our results have demonstrated that including less vulnerable groups, like children, can reduce the impact of vaccine hesitancy for current vaccination campaigns. These results further support the idea of the indirect

benefits of vaccination, which are necessary to achieve herd immunity[7,19]. However, the control of the pandemic as reduction of severe cases (i.e., hospitalisations) and mortality, does not only depend on vaccine uptake but vaccine efficacy and stringency levels of NPIs[7,20,21], which we have represented as underlying transmissibility ($R_t$). Our simulations confirm, that vaccination alone is unlikely to control the current pandemic and NPIs still have a large impact on the epidemic trajectories, until sufficient coverage is reached[22]. In a scenario with lower vaccine efficacy and vaccine hesitancy, longer and more stringent NPIs would be required to compensate lower efficacy as higher coverage levels are required to achieve herd immunity[19].

Our model structure allowed us to capture vaccine hesitancy heterogeneity between age groups[9–11] and analyse its effect in current vaccine rollout plans, which are prioritising older individuals. We have shown that even though older age groups have higher vaccine acceptance levels, these groups have higher mortality in a vaccine hesitancy scenario. As our model does not capture differential risk within sub-populations, it was not possible to assess the effect of vaccine hesitancy in other prioritised populations like health care workers. In which high levels of vaccine hesitancy have been reported despite having higher risk of infection[23].

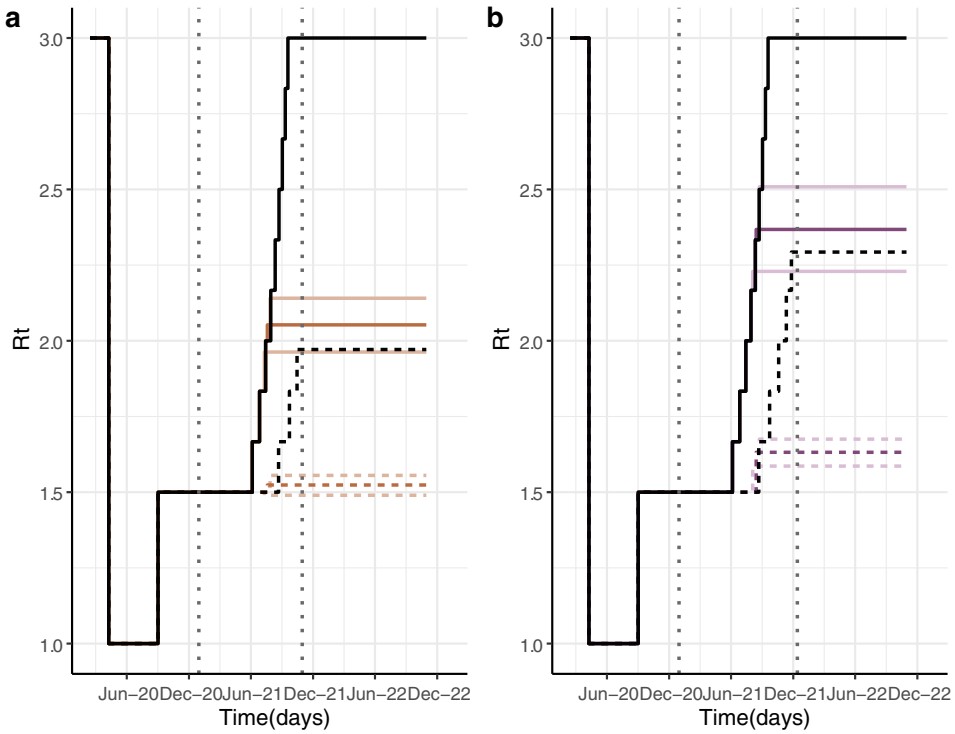

**Fig. 3 Stringency of NPIs required to control the epidemic under different vaccine hesitancy scenarios.** Panel **a** shows Rt profiles for an adults-only vaccination campaign. Panel **b** shows Rt profiles for a vaccination campaign including children. Reproductive number profiles are estimated to keep the herd immunity threshold such that epidemic impact is the same for each scenario as in the ideal scenario. A lower reproductive number corresponds to more stringent NPIs. Continuous lines represent profiles for a high efficacy vaccine and dashed lines represent profiles for a moderate efficacy vaccine. Vertical dotted lines show the period of vaccination in the ideal scenario.

Country fitting showed a higher initial Rt compared to our illustrative example. These values are consistent with those estimated for other European countries, where initial Rt values have been estimated as high as ~4.5, which may be due to possible under-ascertainment in deaths in early periods of the pandemics[1]. It is still unknown how transmission levels will develop in the long term as more transmissible variants are emerging and NPIs behaviour may persist after the pandemic. Here we have assumed a staged release of NPIs with a step-wise increase of Rt, representing governments' easing of restrictions. This step function is a simplification to illustrate the process of balancing the relaxation of NPIs whilst continuing to suppress transmission. Nonetheless, the evaluation approaches introduced in this study can be adjusted to include complex Rt dynamics as more information on COVID-19 transmissibility evolution become available.

Our analysis necessarily makes many simplifying assumptions, and it is important to note that the future trajectory of the epidemic will depend on the complex interactions between vaccination uptake, behaviour and government interventions. First, we have assumed homogenous mixing between vaccine-hesitant individuals. However, as has been seen for other diseases, COVID-19 vaccine hesitancy is heterogenous and clustered within population subgroups[24]. Transmission is more likely to be sustained within clusters with low vaccine coverage[25,26] and therefore future outbreaks may be limited to these sub-populations. Secondly, we have modelled hesitancy levels constant over the time frame analysed; yet, self-reported attitudes to COVID-19 vaccines are changing over time[9,10] as the perceived risk for both disease and vaccines keeps varying[12,21]. Thirdly, we have assumed vaccination rate remains constant over the vaccination period. However, vaccination logistics depend on multidisciplinary factors[27] and both vaccine availably and uptake can be dynamic. Finally, our model does not account for immune escape from the vaccine due to new variants arising. Whilst second-generation vaccines will likely become available to address this issue, it is currently unclear whether some of the high levels of vaccine uptake observed in early vaccine rollouts would be sustained in subsequent booster programmes.

Getting vaccinated is an individual choice, but these individual choices have population wide effects that are likely to challenge current efforts to control COVID-19. Our findings suggest that vaccine hesitancy may have a substantial impact on the pandemic trajectory, deaths and hospitalisation. To prevent such adverse outcomes, NPIs would need to stay in place longer, or possibly indefinitely, resulting in high economic and social costs[28,29]. Reducing vaccine hesitancy is therefore an important public health priority. Interventions that aim to build trust, for example with community-based public education or via positive role-models, are proven efficacious approaches to address hesitancy[30]. There is an ongoing debate about vaccine passports as a condition to travel, or a vaccination requirement for employees[31]. Such interventions may be effective because they incentivize individuals to get vaccinated, but they are controversial in libertarian democracies because they curtail personal freedom and individual choice about medical treatments. The alternative will be to accept some level of disease, hospitalisation and deaths given the level of vaccine coverage achieved whilst allowing NPIs to be lifted, given that NPIs are not a sustainable long-term method for control.

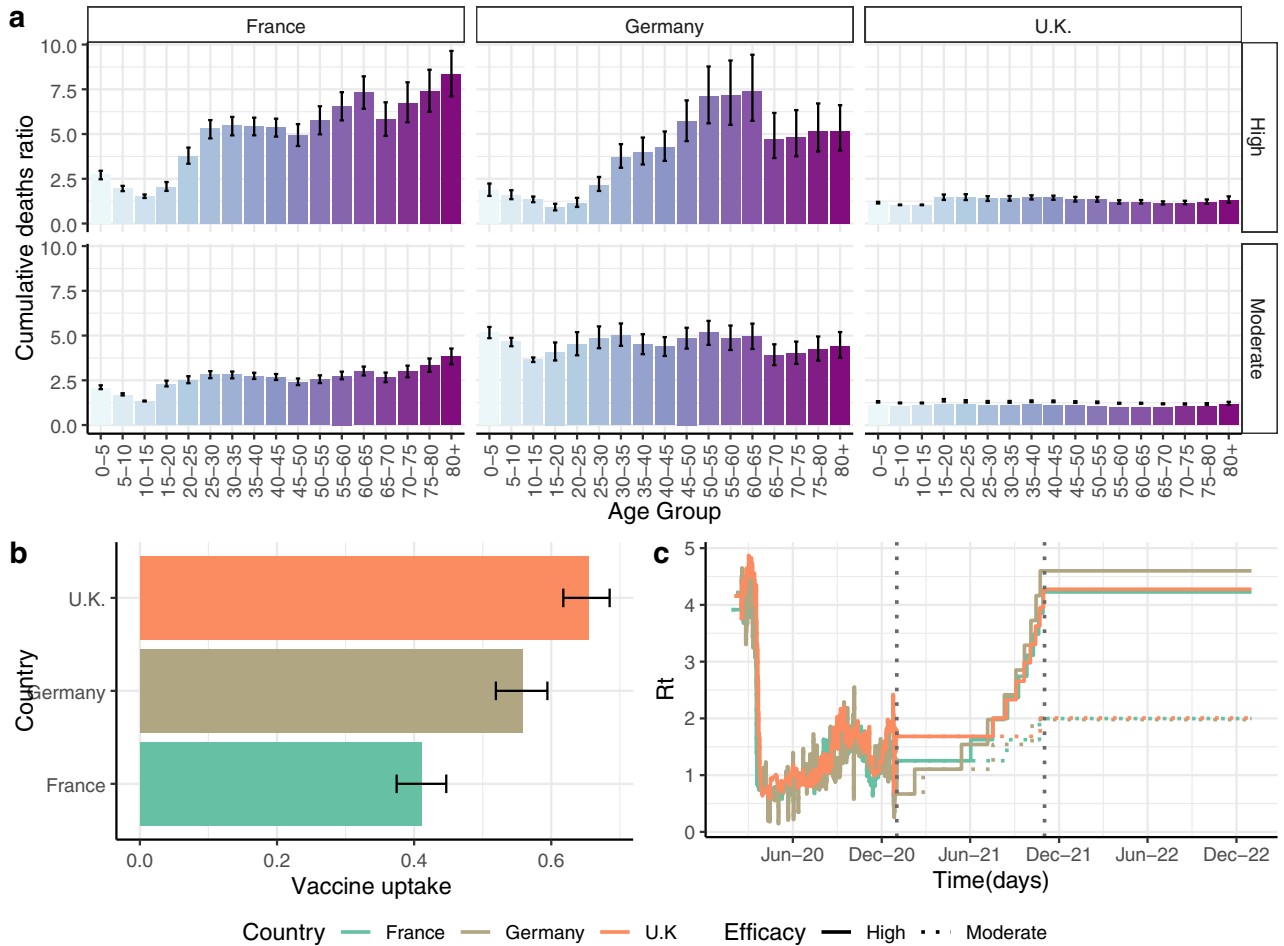

**Fig. 4 Impact of vaccine hesitancy for three European countries. a** Cumulative death ratios per age group compared to the ideal vaccine uptake scenario, by country and vaccine efficacy profile. The ratio compares cumulative deaths projected over a 2-year period after vaccination starts for two scenarios: an ideal scenario, where 95% of the population older than 15 years gets vaccinated and a vaccine hesitancy scenario, where coverage for people over 15 years old is based on vaccine acceptance from b. **b** Reported vaccine acceptance per age group in France, Germany and the United Kingdom reproduced from Jones et al[10]. Values show median vaccine coverage and bars show 10–90% quantiles obtained by running the model at the quantiles from the data. **c** Reproductive number profile for country-specific simulations. Profiles, before vaccination begins, are taken from model fittings to country-specific data (https://mrc-ide.github.io/global-lmic-reports/). After vaccination starts, NPIs are lifted based on an ideal vaccination coverage over time. Reproductive number is set to increase in ten steps from the value at the beginning of vaccination to an average initial reproductive number. Continuous lines show profiles for a high efficacy vaccine. Dotted lines show profiles for a moderate efficacy vaccine.

## Data availability

All data used in this study are from publicly available sources at the links provided in the main text and references. Vaccine hesitancy surveys are from the Imperial College London YouGov Covid 19 Behaviour Tracker Data Hub (https://github.com/YouGov-Data/covid-19-tracker). For ease of reproducibility of our results, the dataset is also stored in our associated publicly available Github repository[32] so that the modelling outputs can be reproduced without further data manipulation. Demographic information is from the United Nations Population prospects https://population.un.org/wpp/). Mobility data from Google (https://www.google.com/covid19/mobility/). And model fittings to country-specific data are from https://mrc-ide.github.io/global-lmic-reports/results. Source data can be found in the Supplementary Data file.

## Code availability

Analyses were carried out in R 4.0.2. Code for the transmission model and analysis is available on GitHub[32]. COVID-19 vaccination model code is available at https://github.com/mrc-ide/nimue[15].

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

## Acknowledgements
All authors acknowledge funding from the MRC Centre for Global Infectious Disease Analysis (reference MR/R015600/1), jointly funded by the UK Medical Research Council (MRC) and the UK Foreign, Commonwealth & Development Office (FCDO), under the MRC/FCDO Concordat agreement and is also part of the EDCTP2 programme supported by the European Union. K.H. acknowledges funding by Community Jameel. P.W. and A.B.H. acknowledge funding from the Imperial College Research Fellowship scheme. A.C.G., D.O.M. and G.D.C. acknowledge funding from The Wellcome Trust (no. 215143/Z/18/Z to D.O.M.).

## Author contributions
A.C.G., K.H., P.W. and D.O.M. conceived the study. A.B.H., P.W., O.J.W. and G.D.C. developed and coded the model. D.O.M. ran the simulations and undertook the analysis with support from P.W.; O.J.W. parametrised the model to country data. D.O.M. produced the first draft of the manuscript with additional input from P.W., K.H. and A.C.G. All authors approved the final version for submission.

## Competing interests
A.B.H., P.W. and A.C.G. declare consultancy fees from the World Health Organization in relation to modelling COVID-19 vaccine impact in the European region, outside the submitted work. The authors declare no other competing interests.
