## [Peer Review File · Communications Medicine]

This manuscript has been previously reviewed at another Nature Portfolio journal. This document only contains reviewer comments and rebuttal letters for versions considered at Communications Medicine.

Reviewers' comments:

Reviewer #1 (Remarks to the Author):

I find this paper very interesting. The topics addressed, as well as the methodological approach are adequate. The paper is fluently written and clear. I feel it is suitable for publication as it is.

Reviewer #2 (Remarks to the Author):

The manuscript studies the effect of vaccine hesitancy on the time evolution of the ongoing COVID-19 pandemic.

It is well written, clear and pleasant to read, accessible to a broad audience.

The conclusions - higher vaccine hesitancy, i.e. lower vaccine uptake, leads to more deaths and potentially to the need to keep NPIs enforced - are not surprising, but the manuscript has interest because it quantifies the effect with a sound modelling approach.

The concerns raised in the previous review round have been addressed satisfactorily in my view.

Concerning comment 6 by Reviewer #1 and comment 1 by Reviewer #2 about the counterintuitive difference between deaths projection with high vs. moderate efficacy vaccines, which is explained by the different stringency of NPIs assumed in the two scenarios: it would be worth pointing out that such an outcome in this manuscript is in full agreement with the claim in the article "Modeling vaccination rollouts, SARS-CoV-2 variants and the requirement for non-pharmaceutical interventions in Italy", Nature Medicine 2021

<https://www.nature.com/articles/s41591-021-01334-5>

stating that NPIs have a higher effect on the epidemic evolution than vaccination alone, hence NPIs need to be kept in place at least during the first phase of the vaccination campaign, until sufficient coverage is reached - and, as shown in this manuscript, sufficient coverage may never be reached due to vaccine hesitancy.

Reviewer #3 (Remarks to the Author):

Overview

In their work, Olivera-Mesa and coauthors aim to study the effect of vaccine hesitancy in the course of the ongoing pandemic. Compared with the version we reviewed for another journal, the manuscript now incorporates the potential impact of vaccinating children, and the narrative and figures have improved. However, major points regarding the modeling aspects of the work (which we list below) remain open and require revision.

Major

General

Parameter choice and the way vaccine rollout is modeled closely resemble the vaccination programs

in the UK, which is markedly different from that of the EU and other countries worldwide. Furthermore, no data-driven analysis or inference is carried out for the vaccine rollout timeframe; thus, the contribution is a modeling study—both title and abstract need to reflect this.

Other aspects that limit the generality of the results relate to the vaccine delivery schedule, as i) age prioritization by bins of 5 years, ii) constant vaccine delivery (not reported) so that all individuals willing to be vaccinated can obtain it within ten months. Furthermore, if we understood correctly lines 258--259, the vaccination period could be much shorter as the vaccination rate is set for the `_ideal_` scenario where 100% of the population aged 15+ get vaccinated within ten months. iii) hospital parameters are not age-dependent and vary from country to country (i.e., the definition of ICU and ICU availability -- which is also not modeled by any means). In countries with fewer available ICU units, prioritization might occur, and not everybody requiring intensive care would have the chance to get it.

It is unclear if, model-wise, vaccine hesitancy implies something different from solely varying the vaccine uptake. In that case, the main findings (more hesitancy --fewer people being protected--, more mortality and morbidity, less freedom) were already reported (e.g., see references below, 1--7). In addition, as mentioned earlier, there are aspects of the model that are original but not described. Explaining those contributions would be crucial for assessing the novelties of the manuscript. Saying it differently, one of the corner stones of assuring scientific quality is reproducibility. Currently, one does not have sufficient information to reproduce the model.

Model

1. Please include in the supplementary materials model equations, so the manuscript is self-contained. New compartments or contributions (as waning immunity and vaccination programs) and the reasoning behind their modeling should be described in detail.

2. Regarding vaccine uptake and vaccine hesitancy, please clarify the following points:

* What is the vaccine uptake selected for each group? i) was it a constant? ii) if not, and it followed the distributions presented in Fig S2, how did the authors interpolate the 5-ages binning they consider for vaccine prioritization (lines 267--270)? How does the model sample from these uptakes, and what is precisely the numerical experiment carried out for quantitative assessment of the impact of vaccine hesitancy?

* Do hesitant individuals have the chance to change their minds and get vaccinated? If not, they would not be "vaccine-hesitant" but rather reject the vaccine when offered.

3. The definition of the "ideal scenario" seems to be rather utopic, and this also increases the relative effect of the variables studied herein. I wonder if the authors could refer their findings to a base uptake (w/o hesitancy) as that reported by Wouters et al. [8], and to explore the hesitancy as deviations from that baseline?

4. Regarding Figure S1 (model)

* Why is there not a connection between hospitalization and ICU? Typically, individuals are first admitted and then eventually require intensive care

* What is the difference between states v_1 and v_2 (and v_3 - v_4 ? What are the transition rates, and how is the coupling performed in the differential equation so that the "peak" in transit from one compartment to another is not at $t=0$?

* Why is there a v_5 state if the authors assume that vaccine protection lasts forever? (Table S1)

5. Regarding vaccine hesitancy:

* What is the percentage of each age group assumed to be hesitant? Please consider reporting it as

a table, as it is a crucial parameter.

* How is hesitancy modeled? Would the authors please clarify whether it is a variation in vaccine uptake and whether hesitant individuals are rather "vaccine-rejectant" individuals?

6. Regarding the reproduction number R_t

* How is this parameter modeled and involved in the equations?

* How does it relate to the basic reproduction number?

* Why does R_t surpass the assumed R_0 in Figure 4 c? In settings where vaccination has already covered a considerable fraction of the population, still having a disease propagating might lead to unfeasible solutions (such as those featuring reproduction numbers larger than the basic one). Worth mentioning, R_0 should not be a property of a given country but rather of the predominant variant.

* Figure 3: sharp transitions in R_t do not seem to be realistic.

7. Regarding waning immunity

* Please include the reasoning behind this contribution in the model (both formulation and differential equation).

* If the contribution is modeled as first-order kinetics (as we can infer from the supplementary materials of ref. 7), what is the percentage of people that already lost their naturally acquired immunity after eight months, if using $\rho=1/365$? Our back-of-the-envelope calculation leads to $\approx 50\%$ (even though eight months was mentioned to be the minimum time for protection to last)

8. Regarding scenario choices

* Why is the analysis window two years long (Figs 1--4)? And how realistic is it considering potentially emerging variants with partial immune escape?

9. Regarding Vaccine logistics, can the authors make explicit the term $\kappa(a)$ listed in Table S1 to know how fast we vaccinate?. How does vaccine availability resemble the increasing stocks?. Can individuals get infected while developing antibodies? Does everybody getting the first dose get the second? Given the broadly reported secondary effects, would not that affect the decision of getting a second dose (and thus impact hesitancy)?

10. Regarding age stratification:

* Please report age groups, sizes, and age-dependent parameters in separate tables.

* Please report the interpolation mechanism for those parameters that are reported with a different age resolution (e.g., vaccine uptake)

References

[1] Moore, S., Hill, E. M., Dyson, L., Tildesley, M., & Keeling, M. J. (2020). Modelling optimal vaccination strategy for SARS-CoV-2 in the UK. medRxiv.

[2] Moore, S., Hill, E. M., Tildesley, M. J., Dyson, L., & Keeling, M. J. (2021). Vaccination and non-pharmaceutical interventions for COVID-19: a mathematical modelling study. The Lancet Infectious Diseases.

[3] Bonsall, M., Huntingford, C., & Rawson, T. (2021). Optimal time to return to normality: parallel use of COVID-19 vaccines and circuit breakers. medRxiv.

[4] Bauer, S., Contreras, S., Dehning, J., Linden, M., Iftekhar, E., Mohr, S. B., ... & Priesemann, V. (2021). Relaxing restrictions at the pace of vaccination increases freedom and guards against further

COVID-19 waves in Europe. arXiv preprint arXiv:2103.06228.

[5] Bubar, K. M., Reinholt, K., Kissler, S. M., Lipsitch, M., Cobey, S., Grad, Y. H., & Larremore, D. B. (2021). Model-informed COVID-19 vaccine prioritization strategies by age and serostatus. *Science*, 371(6532), 916-921.

[6] Wouters, O. J., Shadlen, K. C., Salcher-Konrad, M., Pollard, A. J., Larson, H. J., Teerawattananon, Y., & Jit, M. (2021). Challenges in ensuring global access to COVID-19 vaccines: production, affordability, allocation, and deployment. *The Lancet*.

[7] Viana, J., van Dorp, C. H., Nunes, A., Gomes, M. C., van Boven, M., Kretzschmar, M. E., ... & Rozhnova, G. (2021). Controlling the pandemic during the SARS-CoV-2 vaccination rollout. *Nature Communications*, 12(1), 1-15.

[8] Petherick, A., Goldszmidt, R. G., Andrade, E. B., Furst, R., Pott, A., & Wood, A. (2021). A worldwide assessment of COVID-19 pandemic-policy fatigue. Available at SSRN 3774252. This paper might be interesting about your project.

**Imperial College
London**

**MRC Centre for Global Infectious
Disease Analysis**

Dept. of Infectious Disease Epidemiology
School of Public Health
Imperial College London
St Mary's Campus
Norfolk Place, London W2 1PG
Tel: +44 7 465 688 558

d.olivera-mesa17@imperial.ac.uk

Daniela Olivera Mesa

Dear Reviewers,

Thank you for your taking the time to provide comments and suggestions and the opportunity to revise our manuscript, "Modelling the impact of vaccine hesitancy in prolonging the need for Non-Pharmaceutical Interventions to control the COVID-19 pandemic." to Communications Medicine. We provide a detailed response to each comment in the following.

Reviewers' comments:

Reviewer #1:

I find this paper very interesting. The topics addressed, as well as the methodological approach are adequate. The paper is fluently written and clear. I feel it is suitable for publication as it is.

Reviewer #2:

The manuscript studies the effect of vaccine hesitancy on the time evolution of the ongoing COVID-19 pandemic. It is well written, clear and pleasant to read, accessible to a broad audience.

The conclusions - higher vaccine hesitancy, i.e. lower vaccine uptake, leads to more deaths and potentially to the need to keep NPIs enforced - are not surprising, but the manuscript has interest because it quantifies the effect with a sound modelling approach.

The concerns raised in the previous review round have been addressed satisfactorily in my view.

Concerning comment 6 by Reviewer #1 and comment 1 by Reviewer #2 about the counterintuitive difference between deaths projection with high vs. moderate efficacy vaccines, which is explained by the different stringency of NPIs assumed in the two scenarios: it would be worth pointing out that such an outcome in this manuscript is in full agreement with the claim in the article "Modeling vaccination rollouts, SARS-CoV-2 variants and the requirement for non-pharmaceutical interventions in Italy", Nature Medicine 2021

<https://www.nature.com/articles/s41591-021-01334-5>

stating that NPIs have a higher effect on the epidemic evolution than vaccination alone, hence NPIs

need to be kept in place at least during the first phase of the vaccination campaign, until sufficient coverage is reached - and, as shown in this manuscript, sufficient coverage may never be reached due to vaccine hesitancy.

Many thanks for this interesting and relevant reference which we now refer to in the discussion (Lines 154-156)

Reviewer #3:

Overview

In their work, Olivera-Mesa and coauthors aim to study the effect of vaccine hesitancy in the course of the ongoing pandemic. Compared with the version we reviewed for another journal, the manuscript now incorporates the potential impact of vaccinating children, and the narrative and figures have improved. However, major points regarding the modelling aspects of the work (which we list below) remain open and require revision.

Many thanks to the reviewer for their comments. We used a previously developed mathematical model for our simulations. A description of this model has now been included in the supplementary material. In order to stay within the scope of the manuscript as well as to ensure its reproducibility our revised manuscript also details the parameters for each of the scenarios. The original model has been thoroughly described in reference 2 by Hogan et al. (2021) and the code is openly available from: <https://mrc-ide.github.io/nimue/>

Reference 2: Hogan, A.B., et al. Within-country age-based prioritisation, global allocation, and public health impact of a vaccine against SARS-CoV-2: A mathematical modelling analysis. Vaccine (2021).

Major

General

Parameter choice and the way vaccine rollout is modeled closely resemble the vaccination programs in the UK, which is markedly different from that of the EU and other countries worldwide. Furthermore, no data-driven analysis or inference is carried out for the vaccine rollout timeframe; thus, the contribution is a modeling study—both title and abstract need to reflect this.

Thank you for your comment. At the time of submission, our simulations considered a rollout plan representative of a country that at the time of analysis was in an advanced stage (i.e., England and Israel). This rollout prioritises older age groups and is aligned with WHO recommendations [1] with similar aged-based prioritisation subsequently adopted by many countries worldwide. This is described in lines 274-277 of the methods section: “... *prioritising older age groups: with 80+ years vaccinated first and then sequentially including additional age groups in 5-year age-bands down to 15-19 years for adults only vaccination simulations and down to 5-10 years for simulations including children vaccination*”.

We agree with the reviewer that rollout end dates may differ within countries. However, as the aim of this study is to analyse the effect of vaccine hesitancy across different scenarios, we felt that vaccine rollout should remain constant for all simulations to isolate estimates of the impact of hesitancy specifically.

We agree that this is a modelling study using previously published data and have amended the title and abstract as suggested.

Other aspects that limit the generality of the results relate to the vaccine delivery schedule, as i) age prioritization by bins of 5 years, ii) constant vaccine delivery (not reported) so that all individuals willing to be vaccinated can obtain it within ten months. Furthermore, if we understood correctly lines 258--259, the vaccination period could be much shorter as the vaccination rate is set for the _ideal_ scenario where 100% of the population aged 15+ get vaccinated within ten months. iii) hospital parameters are not age-dependent and vary from country to country (i.e., the definition of ICU and ICU availability -- which is also not modeled by any means). In countries with fewer available ICU units, prioritization might occur, and not everybody requiring intensive care would have the chance to get it.

Thanks for the comment. We understand these points might have caused some confusion.

As noted above, we standardised the roll-out of vaccines in our simulation to prioritise older age-groups. This is in line with WHO recommendations. Whilst countries have opened up across a range of age-groups, the majority have prioritised the oldest first alongside at-risk groups. By standardising the roll-out across simulations using 5-year age bands, we are better able to isolate the impact of vaccine hesitancy on the epidemic. For population-level vaccine roll-out in the timeframes modelled, age-prioritisation in finer or broader age groups would not substantially impact the key findings of this analysis.

In our methods under the parameters section, we describe how a constant vaccination rate was implemented for the different populations modelled: *“We assumed a constant vaccination rate (κ), at which all individuals aged 15 years and above (~78% of the population) will be vaccinated over a 10-month period”*. Since we assume that the vaccination rate is constrained not by vaccine uptake but by the supply and delivery of vaccines, lower levels of coverage result in shorter vaccination campaigns, as pointed out by the reviewer. This is clarified in the methods and further details have been provided in the supplementary material:

“Vaccines are distributed by age groups at a rate κ and a matrix of coverage targets that represents prioritisation strategies (T_{sa}). In this matrix rows (s) represent ordered prioritisation steps and columns (a) the age group. Target coverage per age group was changed according to the different scenarios modelled...”

The model includes age- and country-specific hospitalisation parameters. This has now been clarified in the supplementary material. The parameters are also provided in the supplementary material.

- *“Following infection, individuals in the I_{CASE} state proceed to an ICU unit (I_{ICU}) or hospitalisation in general ward (I_{HOSP}) at rates based on age specific probabilities and hospital bed capacity (...) Additional constraints are included in the hospitalisation pathway to capture situations in which the need exceeds capacity; with those that do not receive appropriate care experiencing higher death rates. These constraints are country-dependent for the country-specific scenarios and are described in Table S1.”*

It is unclear if, model-wise, vaccine hesitancy implies something different from solely varying the vaccine uptake. In that case, the main findings (more hesitancy --fewer people being protected--, more mortality and morbidity, less freedom) were already reported (e.g., see references below, 1--7).

The reviewer is correct that we are modelling the impact of vaccine hesitancy only via vaccine coverage. Nonetheless, we think the evaluation approaches presented in this study are valuable. The unique aspect here is that we explore the impact of age-dependant vaccine hesitancy using age- and country-specific survey data to give plausible bounds on its impact. Given that vaccination rollout is still ongoing in most countries we consider our results significant to reiterate the need for interventions to minimise vaccine hesitancy.

In addition, as mentioned earlier, there are aspects of the model that are original but not described. Explaining those contributions would be crucial for assessing the novelties of the manuscript. Saying it differently, one of the corner stones of assuring scientific quality is reproducibility. Currently, one does not have sufficient information to reproduce the model.

We apologise for omitting some of the relevant detail. As noted below, we have thoroughly revised the supplementary material to include all the relevant detail. We have also included a link to the code on Github so that the analyses are reproducible.

Model

1. Please include in the supplementary materials model equations, so the manuscript is self-contained. New compartments or contributions (as waning immunity and vaccination programs) and the reasoning behind their modeling should be described in detail.

We are using a model previously described by Hogan et al. in [2]. All mathematical details are explained completely in the supplementary information of this reference. We have reproduced the equations into our supplementary material to make the manuscript self-contained.

2. Regarding vaccine uptake and vaccine hesitancy, please clarify the following points:

** What is the vaccine uptake selected for each group? i) was it a constant? ii) if not, and it followed the distributions presented in Fig S2, how did the authors interpolate the 5-ages binning they consider for vaccine prioritization (lines 267--270)? How does the model sample from these uptakes, and what is precisely the numerical experiment carried out for quantitative assessment of the impact of vaccine hesitancy? * Do hesitant individuals have the chance to change their minds and get vaccinated? If not, they would not be "vaccine-hesitant" but rather reject the vaccine when offered.*

Thank you for highlighting these areas of confusion. In order to clarify vaccine uptake per age group we have now included lines 59-60 in the supplementary material:

"Target coverage per age group (C_a) was changed according to the different scenarios modelled (e.g., ideal vs vaccine hesitancy scenarios)..."

Which complement the Scenarios (Lines 300-305) section in the main text:

"We consider two potential scenarios for vaccine coverage target per age group: An ideal scenario where final coverage was 95% for all age groups vaccinated, and a vaccine hesitancy scenario where final coverage per age groups was given by the multinomial distribution from the survey. For the latter scenario, we assumed vaccine hesitancy remain the same within the age bins reported by the survey and we modelled the median coverage per age group as well as the 10% and 90% quantiles, to determine upper and lower bounds."

Survey data was translated to vaccine uptake per age group as described in the methods section (Lines 213-229). For the simulations, this uptake was translated into 5 age bins assuming the level of

vaccine hesitancy remains constant across ages within the same age bin. We have now included Table S3 to make this clearer; we have also updated Figure S2 to highlight the median, 10% and 90% quantiles values used as coverage for the simulations.

3. The definition of the "ideal scenario" seems to be rather utopic, and this also increases the relative effect of the variables studied herein. I wonder if the authors could refer their findings to a base uptake (w/o hesitancy) as that reported by Wouters et al. [8], and to explore the hesitancy as deviations from that baseline?

Thanks for this comment. Our "ideal" scenario assumes 95% vaccine coverage for all adults. This value is in line with vaccine acceptance levels reported by a number of countries in Asia (E.g Vietnam, India and China) where vaccine hesitancy levels are low, as shown by Wouter et al. Furthermore, our ideal scenario is based on observed uptake for well-established vaccine campaigns in western countries. Vaccine uptake has shown to be above 95% for certain age-groups in countries including the U.K [3], Portugal, Malta and Denmark [4]. We therefore disagree that this is "utopic" and feel it is the appropriate comparison.

4. Regarding Figure S1 (model)

** Why is there not a connection between hospitalization and ICU? Typically, individuals are first admitted and then eventually require intensive care*

** What is the difference between states v1 and v2 (and v3-v4? What are the transition rates, and how is the coupling performed in the differential equation so that the "peak" in transit from one compartment to another is not at t=0? * Why is there a v5 state if the authors assume that vaccine protection lasts forever? (Table S1)*

Many thanks for these comments.

Given that the period of time spent in a general bed before being admitted to ICU is typically short, plus there is no difference in onward transmission of the pre-ICU versus ICU state, we chose to simplify the structure and ignore this stage. Previous studies [5] have found that the time difference between being admitted to hospital and then the ICU is only a couple of days. Whilst this does mean that we may over-estimate the need for ICU beds, this effect is marginal in the simulations here since health system capacity is rarely over-whelmed in high-income settings due to their ability to implement surge capacity.

Regarding the different vaccination status, as explained in the new description, there are 4 main statuses:

Stages v1 and v2 correspond to vaccinated but not protected and stages v3 and v4 correspond to vaccinated and protected. There is no difference between v1 and v2, or v3 and v4 – the two compartments are there to generate an Erlang distribution rather than exponential to ensure that there is no peak in transit at t=0. The rate of flow from the unvaccinated to vaccinated states is determined by kappa (see above and supplementary material for the parameter values). The other rates of flow are determined by the parameters stated in Table S1. The model used for our

simulations allows for vaccine immunity to wane over time (v5). Nonetheless, for our simulations we assumed vaccine protection is long-lasting and hence we set the mean duration of vaccine protection to be long (see Table S1).

5. Regarding vaccine hesitancy:

** What is the percentage of each age group assumed to be hesitant? Please consider reporting it as a table, as it is a crucial parameter. * How is hesitancy modeled? Would the authors please clarify whether it is a variation in vaccine uptake and whether hesitant individuals are rather "vaccine-rejectant" individuals?*

In our simulations we modelled vaccine uptake per age group as coverage achieved at the end of vaccination campaign. To estimated vaccine uptake, we translated the survey results as explained in the methods section lines 213-229.

In order to show vaccine uptake clearer, we have included Table S3 and updated Figure S2 to highlight the median, 10% and 90% values for vaccine uptake per age group.

6. Regarding the reproduction number R_t

** How is this parameter modeled and involved in the equations?*

** How does it relate to the basic reproduction number?*

** Why does R_t surpass the assumed R_0 in Figure 4 c? In settings where vaccination has already covered a considerable fraction of the population, still having a disease propagating might lead to unfeasible solutions (such as those featuring reproduction numbers larger than the basic one). Worth mentioning, R_0 should not be a property of a given country but rather of the predominant variant. * Figure 3: sharp transitions in R_t do not seem to be realistic.*

R_t is the current reproductive number. It reflects reductions to the basic reproduction number implied by the combined effect of all non-pharmaceutical interventions currently in place. This value is used to estimate the transmission probability (β). We have included lines 38-43 on the supplementary materials, to include this:

"The level of transmission in the model is parameterised by the reproduction number, R_t , in the absence of vaccine or naturally induced immunity. This is equal to R_0 at the start of the simulation and may be modified forwards in time by the introduction of non-pharmaceutical interventions (NPIs). The transmission probability is obtained as the ratio between the reproductive number and the leading eigen value of the next generation matrix, which depends on duration of infectiousness, the age-stratified mixing matrix and age-dependent probability of hospitalisation."

In the country-specific analyses we used a fitted R_0 at baseline (Figure 4C). We apologise that this was not clear. This has now been clarified in line 320 and the fitted values given in Table S1. The differences between these values and the assumed $R_0=3$ are discussed in lines 168-171 of the manuscript.

Finally, we understand that a step function might not be a realistic representation of an R_t profile. However, given that most countries have gradually eased restrictions in a step-wise manner, we feel that this implementation illustrates the process of balancing the relaxation of NPIs whilst continuing to suppress transmission. Lines 173-177 of discussion now reflect this limitation.

7. Regarding waning immunity

** Please include the reasoning behind this contribution in the model (both formulation and differential equation). * If the contribution is modeled as first-order kinetics (as we can infer from the supplementary materials of ref. 7), what is the percentage of people that already lost their naturally acquired immunity after eight months, if using $\rho=1/365$? Our back-of-the-envelope calculation leads to $\approx 50\%$ (even though eight months was mentioned to be the minimum time for protection to last)*

We apologise for this confusion. The immune state R is split into two compartments, R1 and R2, in order to generate an Erlang distribution. Under this distribution, the loss occurs substantially later than for an exponential distribution as the reviewer points out. We have clarified this in the manuscript's model description (lines 237-238):

“Waning immunity is captured by recovered individuals returning to the susceptible compartment following an Erlang distribution”

8. Regarding scenario choices

** Why is the analysis window two years long (Figs 1--4)? And how realistic is it considering potentially emerging variants with partial immune escape?*

Many thanks for this comment. We selected our simulation time window such that the impact of the initial vaccination response could be evaluated balancing out against future uncertainties, which would increase with a longer time horizon. We feel that two years is a reasonable balance. We are aware of the importance of potentially emerging variants that might affect our simulation results. However, we felt it beyond the scope of the current study to address this specifically. The discussion of the study limitations addresses this point.

9. Regarding Vaccine logistics, can the authors make explicit the term $\kappa(a)$ listed in Table S1 to know how fast we vaccinate?. How does vaccine availability resemble the increasing stocks?. Can individuals get infected while developing antibodies? Does everybody getting the first dose get the second? Given the broadly reported secondary effects, would not that affect the decision of getting a second dose (and thus impact hesitancy)?

i) We have added κ values for each scenario in Table S1.

ii) As noted above, we assume that the rate of vaccination is constant and hence this rate reflects both the supply and the health system delivery.

iii) We assume that those in the v1 and v2 states can still be infected whilst they are developing antibodies. This has been clarified in the text in the supplementary material and is clear in the equations.

iv) We do not explicitly model two separate doses – rather we ignore the early efficacy of the vaccine and assume full protection 28 days after dose 1 (assuming that dose 2 is given at this time). This is now clarified in the supplementary material. In effect, this means that our measure of vaccine hesitancy assumes that those that refuse to be vaccinated do not get either dose.

10. Regarding age stratification:

* Please report age groups, sizes, and age-dependent parameters in separate tables.

* Please report the interpolation mechanism for those parameters that are reported with a different age resolution (e.g., vaccine uptake)

The model is discretised in 17 age categories by bins of 5 years starting at 0-4 and going up to 80+ years old. The size of each age category depends on the demographics from each country modelled (e.g. U.K demographics for the representative scenario).

Regarding the age specific parameters, we have included a Tabel S2 in supplementary material with all age dependant hospitalisation parameters, Table S3 with age specific vaccine uptake and Table S4 with country demographics.

Finally, as noted above in point 2, survey data was translated to vaccine uptake per age group as described in the methods section (Lines 213-229).

We think that the changes following reviewers' comments have improved the manuscript and we thank the reviewers for these. We look forward to hearing from you.

Yours sincerely,

Daniela Olivera Mesa

References

1. World Health Organisation (WHO). *WHO SAGE Roadmap For Prioritizing Uses Of COVID-19 Vaccines In The Context Of Limited Supply*. 2021 [cited 2021 22 September]; Available from: <https://www.who.int/publications/i/item/who-sage-roadmap-for-prioritizing-uses-of-covid-19-vaccines-in-the-context-of-limited-supply>.
2. Hogan, A.B., et al., *Within-country age-based prioritisation, global allocation, and public health impact of a vaccine against SARS-CoV-2: A mathematical modelling analysis*. *Vaccine*, 2021.
3. NHS. *COVID-19 Vaccination Statistics: Week ending Sunday 12th September 2021* 2021; Available from: <https://www.england.nhs.uk/statistics/statistical-work-areas/covid-19-vaccinations/>.
4. European Centre for Disease Prevention and Control. *COVID-19 Vaccine Tracker*. 2021 22 September [cited 2021 September 22]; Available from: <https://vaccinetracker.ecdc.europa.eu/public/extensions/COVID-19/vaccine-tracker.html#age-group-tab>
5. Knock, E.S., et al., *Key epidemiological drivers and impact of interventions in the 2020 SARS-CoV-2 epidemic in England*. *Science Translational Medicine*, 2021. **13**(602): p. eabg4262.

REVIEWERS' COMMENTS:

Reviewer #3 (Remarks to the Author):

We thank the authors for the thoughtful consideration of our comments. All the points we mentioned in the first round of review have been incorporated or answered. We do not have further concerns.